# A Literature Review of Proton Beam Therapy for Prostate Cancer in Japan

**DOI:** 10.3390/jcm8010048

**Published:** 2019-01-05

**Authors:** Rika Maglente Hoshina, Taeko Matsuura, Kikuo Umegaki, Shinichi Shimizu

**Affiliations:** 1Faculty of Medicine and Surgery, University of Santo Tomas, España, Manila 1002, Philippines; 2Proton Beam Therapy Center, Hokkaido University Hospital, Sapporo 060-8648, Japan; matsuura@med.hokudai.ac.jp (T.M.); umegaki@eng.hokudai.ac.jp (K.U.); sshing@med.hokudai.ac.jp (S.S.); 3Global Station for Quantum Medical Science and Engineering, Global Institution for Collaborative Research and Education, Hokkaido University, Sapporo 060-8638, Japan; 4Division of Quantum Science and Engineering, Faculty of Engineering, Hokkaido University, Sapporo 060-8628, Japan; 5Department of Radiation Oncology, Graduate School of Medicine, Hokkaido University, Sapporo 060-8638, Japan

**Keywords:** proton therapy, prostate cancer, literature review, Japan

## Abstract

**Aim:** Patients of proton beam therapy (PBT) for prostate cancer had been continuously growing in number due to its promising characteristics of high dose distribution in the tumor target and a sharp distal fall-off. Considering the large number of proton beam facilities in Japan, the further increase of patients undergoing this treatment is due to the emendations by Japanese National Health Insurance (NHI) and the development of medical equipment and technology, it is necessary to know what kind of research and advancements has been done on proton therapy for prostate cancer in the country. For these reasons, this literature review was conducted. The aim of this review is to identify and discuss research studies of proton beam therapy for prostate cancer in Japan. These include observational, interventional, and secondary data analysis of published articles. **Method:** A literature review on published works related to proton beam therapy for prostate cancer in Japan was conducted using articles that were gathered in the PubMed database of June 2018. We went through abstracts and manuscripts written in English with the keywords ‘proton beam therapy’, ‘prostate cancer’, and ‘Japan’. **Results:** A total of 23 articles were included. Fourteen articles were observational studies, most of which focused on the adverse effects of Proton Beam Therapy (PBT). Seven articles were interventional studies related on treatment planning, equipment parts, as well as target positioning. Two were secondary data analysis. The included studies were published in 13 different journals by different institutions using various equipment. **Conclusion:** Despite the favorable results of proton beam therapy, future research should include more patients and longer follow-up schedules to clarify the definitive role of PBT, yet, up to recent retrospective studies, included in this paper, concluded that PBT can be a suitable treatment option for localized prostate cancer. In addition, interventional studies were conducted by several institutions to further embellish proton therapy.

## 1. Introduction

The prostate is a biologic structure found only in males. It is a walnut-shaped organ that increases in size as one age. The prostate is situated as inferior to the bladder and anterior to the rectum and seminal vesicles. The prostatic part of the urethra passes through the center of this organ that acts as a passageway for urine out of the body. The prostate secretes fluid, which is a part of the composition of the semen, is in charge of providing nourishment and protection to the sperm.

According to Cancer Research UK, there is an estimated 14.1 million new cases of cancer globally. The top four cancers occurring worldwide are lung, breast, bowel, and prostate cancer, respectively. Prostate cancer (PCa) is due to the uncontrollable growth of cells in the prostate gland. In men, prostate neoplasm is the most common form of non-skin cancer, and fortunately, only one out of 35 individuals affected die from this disease due different medical interventions offered to treat prostate cancer [1].

Prostate cancer gradually develops, and depending on the age of diagnosis, patients may have other co-morbidities. Therefore, individualized treatment modalities should be considered as whichever is the most suitable in each case. PCa has a variety of management options such as surgery, radiotherapy, hormone therapy, ablative therapy, active surveillance, as well as multimodal therapies. Active surveillance or active monitoring is used in low-risk and intermediate-risk PCa with low tumor volume. It is performed using prostate-specific antigen (PSA) and annual biopsies to establish disease progression; if this happens, other treatment options are presented. Regardless of the effectiveness of surgery and radiotherapy in cancer eradication, these techniques are associated with sexual and urinary side effects and should be taken into consideration when preferred as the treatment option. Hormone therapy or Androgen Deprivation Therapy (ADT) is only utilized for PCa patients with high-risk or metastasized disease, despite its beneficial risk-benefit ratio due to its linkage in various somatic side effects, including extensive tissue expression of sex steroid hormones. Along with this, androgen deprivation therapy (ADT) is the most common cause of hypogonadism in the contemporary period, and thus, prescription of this modality should be thoroughly weighted on its advantages and disadvantages. The multi-modal treatment option, such as radiation therapy in combination with ADT, aims to increase positive long-term outcomes through addressing local tumor burden and metastatic diseases. Free ablative therapies such as cryotherapy, high intensity focused ultrasound, electroporation, and photodynamic therapy are used as focal therapies that target the tumor itself, and hence, has less association with adverse effects [2,3,4].

The leading cause of death in Japan is cancer since 1981. In 2016, lung cancer ranked first, followed by stomach cancer then colon cancer. Prostate cancer ranks sixth in men with 11,803 mortalities [5]. According to the World Population Review, in 2017, Japan placed second in world life expectancy following Hong Kong with longevity of 80.98 and 87.14 years for men and women, respectively [6]. However, the prolonged life expectancy also increases the probability of cancer. Today, two of three Japanese citizens are diagnosed with the said disease. Since overcoming cancer is directly linked to being able to further raise the average life expectancy in Japan, the Japanese government had put in efforts on funding advanced research in cancer, with one being proton beam therapy (PBT) [7].

One of the standard treatments for localized prostate cancer is radiation therapy, but its considerable acute and late adverse effects to the gastrointestinal (GI) and genitourinary (GU) tract have been a major concern for both patients and the physicians. In a study conducted by Fiorino et al., the risk of experiencing ≥grade 2 GI and GU is about 5%–20% [8]. With the materialization of image guided radiotherapy (IGRT), 3-dimensional conformal radiotherapy (3D-CRT) and intensity modulated radiotherapy (IMRT), the risk of toxicity on the organs at risk (OAR), namely the bladder, rectum, and seminal vesicle were decreased to 5%–10% [9]. The use of proton beams in radiation therapy further reduces this probability with its characteristic Bragg peak, where protons halt at a determined depth, analogous to its energy. The two proton beam delivery techniques- passive scattered or intensity modulate proton therapy localizes the irradiation dose to the organ and decreases the damage to adjacent tissues. Despite the advantage of PBT and the increasing number of patients undergoing the treatment, there are still factors affecting the usage of PBT as the principal standard treatment for localized PCa, namely, the size of the device, cost, PBT facilities, and insurance [10]. However, with the amendment of policies by the Japanese National Health Insurance last April 2018, proton beam therapy for prostate cancer and head and neck cancer are now included in the insurance coverage lowering the financial burden for patients who will undergo this procedure [11]. As a result, it can be predicted that the use of proton beam treatment for localized PCa will continuously increase in the near future.

Since the proposed project in 1973 by High Energy Accelerator Research Organization on the usage of proton beam particles and the actual experimental period on the year 2000 of less than 100 patients, the number of facilities as well as patients had gradually increased annually with around 24,000 patients in 2015 [12]. With the large number of proton beam facilities in the country, the further increase of patients undergoing this treatment due to the emendations of NHI and the development of medical equipment and technology, it is necessary to know what kind of research and advancements has been done on proton therapy of prostate cancer in Japan. For these reasons, this literature review was conducted.

## 2. Results

This literature review used PubMed as its primary and only source for the journals included in this paper. A total of 50 articles were found with the combination of the following key words “Proton Beam Therapy”, “Prostate Cancer”, and “Japan”. Four of the journals which were written in Japanese and one in French were not included. After analysis of the 50 articles, only 23 articles were suited in the literature review that focuses on proton beam therapy for localized prostate cancer in Japan. Articles that focus on other radiation treatments and/or different organs, aside from the prostate, were not entailed in this paper.

Results of the PubMed search showed 23 articles that fit the inclusion criteria for this literature review. These articles were published in 13 different journals. Most of the articles included in this paper had been published in the “International Journal of Radiation Oncology Biology Physics”, followed by “Journal of Radiation Research” and “Medical Physics”. The remaining journals had only one or two articles issued per title (Table 1).

Studies were conducted at different institutions. Hospitals in order of number of published articles are as follows: Five in Hyogo Ion Beam Medical Center, and four in National Cancer Center Hospital East, Shizuoka Cancer Center, Medipolis Proton Therapy, and Fukui Prefectural Hospital. Two articles were published from Hokkaido University Hospital, University of Tsukuba Proton Medical Research Center, Nagoya Proton Therapy Center, and Southern Tohoku Proton Therapy Center. Some of the articles included in this literature review were multi-institutional studies.

Equipment used for treatment planning and beam delivery were predominantly manufactured by Japanese companies such as Hitachi Ltd., Toshiba, and Mitsubishi Electric Corp. Journal titles, institutions, and equipment manufacturers are summarized in Table 1.

Among the publication journals, International Journal of Radiation Oncology Biology Physics, has the most number of published articles and has the highest impact factor (IF) of 5.554. The second journals with the most number of articles were Medical Physics and Journal of Radiation Research have IF of 2.617 and 2.031, respectively. Further impact factors of the other journals can be found in Table 1.

### 2.1. Treatment Planning

Treatment planning is a very important step for obtaining the desired delivery dose for the patient. This step is crucial for the whole treatment and profound execution should be observed. However, potential sources of error inherent in the different types of radiation therapy should be considered, such as precision of patient positioning, anatomical variations, and organ motion. Along with this, there are also disparities in the treatment planning and delivery systems among the proton therapy centers [13]. Several algorithms encompass treatment planning software packages to estimate dose distribution to the organs. First, the uniform-intensity beam algorithm is the simplest and fastest, however, the least accurate due to its inadequate estimation of heterogeneities affecting the beam path. Second, the pencil beam is the generally used algorithm in proton dose calculation. This provides better control as it directly indicates the fluence at any points within the beam, thus, providing a more accurate prediction of the heterogeneities in the proton beam path. The third and the gold standard for modeling dose is the Monte Carlo algorithm. It produces the greatest accuracy in proton dose calculation as it illustrates individual proton interaction with the tissues as it traverses through the patient [14]. However, despite its superiority compared to other algorithms, it is not clinically utilized due to the vast computation time needed. As stated by Paganetti et al., Monte Carlo dose calculation takes about six hours per patient [15]. Kohno et al. developed a graphics processing unit (GPU)-based simplified Monte Carlo (SMC) method and it is compared to a central processing unit (CPU)-based SMC. Clinical implementation of the GPU-SMC was applied to the head and neck, lung, and prostate cancer patients. All the cases resulted in a decreased computation time using the developed GPU-SMC. Specifically, for the prostate case, computation time was six minutes and 25 seconds with CPU-SMC and GPU-SMC, respectively. This study also concluded that calculation volume and proton energy can influence the computation time in Monte Carlo simulations. Hence, the GPU-SMC developed by the researchers demonstrate its successful execution on clinical proton beam planning [16].

### 2.2. Acute Toxicity

Genitourinary and gastrointestinal toxicities are often observed after completion of proton beam therapy. Patient eligibility in these studies include: (i) Pathologically proven adenocarcinoma, (ii) Eastern Cooperative Oncology group performance status, (iii) no serious complications (iv) appropriate organ function (v) no previous history of pelvic radiotherapy, and (vi) written informed consent. Genitourinary toxicity generally involves the bladder with urinary frequency, dysuria, urinary retention, and narrow stream as its common manifestations, while gastrointestinal morbidity is concerned with rectal problems such as proctalgia, soft stool, anal discomfort, and rectal bleeding [17]. In 2005, a multi-institutional study was conducted in 30 patients who had undergone radiotherapy with photons and protons boost. Photon irradiation was given at 50 Gray (Gy)/25 fractions for both the prostate and the bilateral seminal vesicle followed by 26 Gy/13 fractions proton boost therapy to the prostate alone. Acute genitourinary grades 1 and 2 were observed in 20 and four patients, respectively. Grade 2 acute GU toxicity complaints include urinary frequency and urgency. There were also 17 patients who experienced grade 1 acute gastrointestinal toxicity [18] (Table 2). As documented by Khmelevsky et al., a positive decrease in the acute and late grade 2 gastrointestinal morbidity can be observed with the use of additional proton beam boost, as compared to 3D-CRT monotherapy [19].

Mayahara et al. reported 287 patients who underwent proton therapy in Hyogo Ion Beam Medical Center. Ninety-four percent of the patients experienced acute GU morbidity, of which grade 1 toxicity demonstrated minor symptoms and no intervention was given. Grade 2 toxicity was prescribed with medications and grade 3 had distressing symptoms affecting daily life. Of the 111 individuals with grade 2 GU, hematuria, followed by urinary frequency and urinary retention, were the common manifestations of the morbidity. Acute GI was also observed in five patients, which includes either a mild rectal discomfort or a slight increase in bowel movements. Univariate analysis showed that larger clinical target volume (CTV) and old age are significant factors for developing grade 2 and higher GU morbidity. In addition, the multivariate analysis performed showed that greater clinical target volume and the utilization of androgen suppression therapy were significant factors for the prediction of acute Grade 2–3 GU morbidity [20], coinciding with Takagi et al. [25]. Patients who were given neoadjuvant ADT with poor baseline urinary function were 4.3 times more likely to have acute Grade 2 GU complications than those who had neither [20].

The first clinical trial for prostate cancer in 2005, with 30 patients reported grade 2 rectal and bladder acute toxicities present in 0.7% and 12%, respectively. Rectal and bladder toxicities were assessed by incidence of proctitis, rectal bleeding, rectal pain, hematuria, urinary frequency, urgency and retention, as well as dysuria [18]. A study conducted to compare conventional (CPT) and hypofractionated proton therapy (HFPT) was done in 2017 with five hundred and twenty-six patients from February 2013 to May 2016. 74 Gy/37 fractions and 78 GY/39 fractions were irradiated to low and high risk prostate cancer in 254 conventional therapy patients, respectively, while low risk received 60 Gy/20 fractions and high risk had 63 Gy/21 fractions in 272 patients of HFPT. Results in this study, as seen in Table 2, showed that hypofractionation is a more favorable technique in terms of decreasing GU complications than conventional proton therapy, although no significant difference observed between the two in causing GI morbidity. Table 2 shows the results of morbidities in varying EBRTs from completed clinical trials.

The International Prostate Symptom Score (IPSS) is used to assess patient urinary function. Nakajima et al. published IPSS results of HFPT, which were noted to worsen one month after the completion of the treatment. Univariate analysis showed that fractionated schedule, hypertension, National Comprehensive Cancer Network (NCCN) risk group, baseline American Urological Association (AUA) class, and the beam delivery technique are factors associated with the risk of developing acute grade 2 GU toxicity [24]. Arimura et al. conducted a prospective cohort study, which includes 218 individuals, from January 2011–July 2014. 78 Gy, 74 Gy, and 70 Gy were irradiated depending on the risk status of the patient and the cumulative incidence of GU toxicities were as follows: 28.1%, 27.9%, and 13.1% for acute, and 3.4%, 3.5%, and 3.3% for late toxicities [27].

Dermatitis is also one of the noted acute side effects of ongoing proton beam therapy. In 2015, a patient that was given multimodal treatment of proton irradiation and Pirarubicin experienced grade 2 dermatitis and dysuria during and after the treatment. The therapeutic effect was observed on the primary tumor and metastases, which were attributed to proton irradiation and Pirarubicin, respectively [28]. Contact dermatitis was also noted in 46 (18.1%) conventional and 18 (6.6%) hypofractionated patients in a study by Nakajima et al. [24].

### 2.3. Late Toxicity

Late toxicity is observed six-months onwards after completion of the radiation therapy. In a study by Nihei K. et al, late GI and GU toxicities were noted in 16 among 30 participating patients with a median follow-up of 30 months. Two percent had rectal and 4.1% had bladder toxicity among the 151 patients [18]. In 2017, 1375 patients participated in a study on long term outcomes of proton therapy for prostate cancer. GU and GI toxicities of Grade 2 and higher were detected in 2.0% and 3.9%, respectively. (Table 2) Late GU incidence continued to increase while the number of late GI toxicities stabilized five years after the therapy [25]. A retrospective study in 2017 included 93 patients, wherein four of the patients with grade 2 late GU toxicity experienced urinary frequency, while one patient had hematuria. A patient with grade 3 late GU toxicity had non-invasive cystitis. Late grade 2 GI rectal bleeding was also observed in four patients. Analysis in this study showed that the use of anticoagulants is a positive prognostic factor for the development of GI toxicity [9].

According to the study of Iwata et al. involving 1291 patients, long term adverse effect of prostate cancer therapy produced grade 2 GU and GI toxicities as seen in Table 2. Two hundred eighteen patients from a different hospital had consented for another study where Grade 2 and higher GI toxicity was observed in 10.5% of patients who received 78 Gy, 2.3% of 74 Gy, and none of the 70Gy patients [26]. Proton-photon combined treatment adopted by Nihei et al. resulted to two patients with grade 1 and 3 patients with grade 2 late GU morbidity. For GI, 8 experienced grade 1 and 3 patients manifested grade 2 morbidity. Similar modality of proton-photon therapy was also noted to have a decreased late post irradiation rectitis with a reduction rate of about 30% in late grade 2 gastrointestinal complications, as compared to photon irradiation alone [23].

Arimura et al. included sexual score measurement as one of the late adverse effect of the treatment. Using the expanded prostate cancer index composite (EPIC), decreased scores with age was documented, 40-year old patients scored 70 and 30 for 80-year old patients. Data also showed that scores were at their lowest three years after the treatment, followed by a slight recovery. Thus, patient age can be a big factor in measuring EPIC scores after PBT [27].

### 2.4. Biochemical Relapse and Survival Rate

Biochemical relapse or biochemical recurrence is an increase of prostate specific antigen in the blood after completed surgery or radiation [29]. Takagi et al. observed 1375 patients for any biochemical relapse. Five-year biochemical relapse free rates for low risk patients were noted at 99%, intermediate risk at 91%, high risk at 86% and 66% for very high risk. Eight-year biochemical relapse free was also observed at 95%, 87%, 71%, and 55%, respectively. (Table 2) One hundred seventy-seven patients had biochemical relapse with median time of 39 months wherein low and intermediate risk groups had longer median times for biochemical relapse compared to the high and very high risked. Forty-nine of the 177 biochemical recurrence occurred five years or more after the proton therapy. Older patients are less likely to experience biochemical relapse compared to individuals who developed cancer in a young age, which is contrary to expectations. Actually, patients who are less than 64 years old are twice more likely to develop relapse compared to those who are older than 70, hence implying that younger age is a strong prognostic factor favoring biochemical relapse recurrence. Aside from this, the use of anticoagulant and diagnosed diabetes mellitus were also noted to induce incidence. Clinical recurrence was also observed in 43 patients, in which 11 are local recurrences, 15 pelvic lymph nodes, 18 bone metastases, and three others. Cancer Specific Survival rates were as follows: 100% for low and intermediate risks, 99% for high risk, and 95% for very high risk [25].

In a multi-institutional retrospective study including 1291 patients, biochemical relapse occurred to 137 patients wherein 35 of them exhibited clinical relapse. Nine patients had local recurrence, 12 had lymph node metastases, and 17 had bone or lungs metastases. Overall, biochemical relapse free survival was at 97%, 91.1%, and 83.1% for low, intermediate, and high risk patients, while overall survival was 98.4%, 96.8%, and 95.2%. Another study determined the five-year progression free survival rate for intermediate and high risk patients were 97% and 83%, respectively, and overall survival rates measured 96% and 98% [27]. Univariate analysis showed several factors associated with biochemical relapse free survival such as NCCN classification, age, performance status, operability, T stage, Gleason score, PSA value, and ADT. Performance status is an important aspect for both biological relapse free survival and overall survival. Multivariate analysis showed that NCCN classification is a risk factor associated in biochemical relapse free survival, but not on the overall survival of the patient [26].

### 2.5. Passive Scattered Proton Therapy

Passive scattered proton therapy (PSPT) is a proton beam delivery technique that uses a scattering material for proton beam dispersion. A single scatterer can widen the radiation beams adequately for small beams, while a second scatterer is required to establish homogeneous dose profile for larger fields. Collimators and compensators are made accordingly to correspond dose to the target volume. In fact, collimators are the scatterers used in passive scanning. Spread-out Bragg Peak (SOBP) in passive scanning treatment is achieved using range modulator fields or ridge filters located inside the gantry [30]. Murai et al. differentiated two collimators systems, the conventional circular collimator (CC) and the multileaf collimator (MLC). Cases in this study include PBT for brain (5), liver (10), and prostate cancers (10). Fixed planning measurement were used in CC planning; 1 cm targets used 0.5 cm and 0.75 cm for planning target (PTV). The multileaf collimator can be shaped without restrictions thus generating a more adequate beam size to contain the target. Three clinical situations were assessed to compare dose distributions and treatment times between CC and MLC plans on brain, liver, and prostate target volumes. Various determinants were measured in MLC and CC planning, namely homogeneity index, conformity index, minimum and maximum PTV, dose distribution in organs at risk, treatment time, and monitor unit. Results concerning maximum dosage to the rectum, treatment time, and dose to the bladder between the MLC and CC plans showed that the former reduced the first two aspects, while no significant difference was observed in the bladder dose. The use of multileaf collimators in prostate tumors decreased the dose delivery to the rectum, as well as the conformity index compared to circular collimators. This results to a more efficient therapy averting radiation to the normal surrounding tissues [31].

Another study in 2017 had focused on deformable image registration (DIR) uncertainties of dose accumulation in proton therapy. Abe at al. assessed PSPT DIR uncertainties outcome on prostate. DIR determines acquired dose around the target volume and can be evaluated using dice similarity coefficient and surface distance (Hausdorff distance). Dose accumulations on 10 patients who have completed proton beam therapy were analyzed using two DIR software’s, namely Velocity and Raystation. Velocity is an intensity-based DIR that only depends on the image intensity information, while Raystation depends on both image intensity and anatomical information coming from the contoured image. In comparing the two DIR software’s, Raystation is a more accurate program in converting a moving image to stationary ones, due to its high dice similarity coefficient and smaller surface distance. In addition, it presented increased dose-volume histogram (DVH) parameter values for both rectum and bladder. Results from this study indicated that accumulated dose quantities differ depending on which DIR algorithm is used on prostate PSPT [32].

### 2.6. Intensity Modulated Proton Therapy

Intensity modulated proton therapy (IMPT) is the other proton beam delivery technique that makes use of pencil beam scanning that allows larger treatment size and supports increased flexibility in dose-shaping efficiency and dose conformity. A number of hospitals make use of this technique, such as the Hokkaido University Hospital [20].

Matsuura et al. analyzed the distortion effect of the application of gold markers in proton treatment of prostate cancer using the Monte Carlo simulation. Two gold markers sized 2 and 1.5 mm visible on fluoroscopy were used. Three beam directions were utilized for this simulation, right lateral (field 1, ∠270°), left lateral (field 2, ∠90°), and anterior angle (field 3, ∠0°). Biological effects of the dose distortions were analyzed using dose distribution. Improving dose distortion by adding the number of fields outweighs the expanding dose of the shaded region, resulting in an improvement of tumor control probability. Additionally, the marker placed close to the distal part of the beams exhibited minimum dosage, while placing it upstream of the CTV creates downstream dose recovery. Also, this simulation concluded that usage of two or more fields with a marker size of 1.5 mm does not affect the tumor-control probability. However, 2 mm markers require more than two irradiation fields to outweigh the decrease the tumor control probability by less than 3% [32].

A simulation study on the benefit of real time motion compensation was conducted by Fujii et al. in 2017. This simulation study focused on the influence of intrabeam patient repositioning in the real-time image gated proton beam therapy system of 9 prostatic cancer patients which indicated that patient positioning during beam delivery is an effective way to obtain better target coverage and uniformity, while reducing the target margin when the prostate moves during irradiation [33].

### 2.7. Comparison between IMPT and PSPT

Comparison between IMPT and PSPT delivery system was conducted in a clinical study by Kase et al. The efficiency of the two techniques in terms of proton distribution to the tumor and organ at risk were documented. Sixteen patients participated in a study of treatment planning comparison between the techniques, four of whom were diagnosed with prostate cancer. IMPT has a greater ability in concentrating the dose to the patient target volume (PTV) in comparison to PSPT, while the PTV dose homogeneity were noted to be more desirable in PSPT. Decreased dose homogeneity of IMPT was due to hotspots near the border of the PTV, while increased dose homogeneity in PSPT can be credited to the utilization of static ridge filters designed to administer constant dose in the SOBP region and the placement of beam collimator close to the patient’s body surface. This study confirmed that IMPT can reduce large doses to the OAR, yet it was not effective in decreasing the maximum dose irradiation to the skin of prostate cancer patients. Hence, the use of IMPT is not beneficial in reducing the probability of inflammation. However, due to the decreased dosage to the OARs incidence of rectal bleeding and irradiation cystitis is also reduced. The utilization of IMPT can enhance the dose concentration by around 50% of the isodose line. Another significant factor to consider is the beam angle selection in order to produce the optimal treatment outcome when using IMPT [29].

### 2.8. Target Positioning and Organs at Risk

Target positioning is a very important factor to deliver adequate amount of dosage to the target with minimal radiation to the organ at risk. In 2018, Maeda et al. compared bone and prostate matching in terms of the dose constraint in the rectum and dose coverage in the prostate which resulted to more favourable outcomes in the latter. Maeda et al. measured movements of the prostate, seminal vesicles, as well as the rectum at the time of CT image guiding proton therapy for prostate cancer while studying the range difference in lateral opposed proton beams. A total of 375 CT images of 10 patients were taken to assess the movements of the prostate, seminal vesicle, and rectum with the use of bone, prostate center (PC), and prostate-rectum boundary (PRB) matching strategies. Extensive movement of the seminal vesicle towards the superior-inferior direction and anterior region of the rectum was observed. Comparison of the three matching strategies showed that PRB matching exhibited the least positional disparity on all directions specifically in the posterior part of the prostate however, there was no significant difference on the anterior part positioning. The use of PRB matching along with CT guidance image can be more beneficial in decreasing rectal toxicities. It was observed that errors are increased in the anteroposterior (AP) and superoinferior (SI) parts of both the prostate and seminal vesicle, yet a larger deviation was found in the latter alongside its lateral portion. The prostate had shown a higher daily disparity on the AP side than the SI direction indicating that prostate movements are inclined toward the anterior side. In CT image guided proton therapy, keeping the dose constraint of the rectum and the dose coverage of the prostate can be achieved by proper repositioning of AP and SI direction in conventional bone matching. PRB matching showed the least average positional deviation along the AP and SI direction compared to the other two techniques. Bone matching and prostate center matching increased the positional deviations and errors of the rectal wall in the inferior to superior side, but less can be observed in PC. In the case of PRB matching, errors and average positional deviation were minimal around the center of the prostate due to the decreased movement of the seminal vesicle (SV) in this matching strategy. Thus, positional disparity due to SV movements can be better balanced out by PRB matching in comparison to PC and bone matching [34,35].

Fuji et al. conducted a clinical research on the effect of rectal emptying tube (RET) to the rectal volume and prostate localization. Pronounced internal motion of the prostate was observed in the AP direction, agreeing with previous reports [34,35]. The relationship between the reduction of rectal volume and disparity in prostate motion was also confirmed in this study. Internal motion can be influenced by various factors such as rectal filling, bladder filling, leg position, respiratory position, and most importantly, rectal volume. Twenty-one patients had given their consent to use a RET which releases gas from the rectum to mechanically control the its volume. This device is also useful for immobilization of the organ and also hampers rectum shape change after emplacement. Results showed that RET can decrease the rectal volume and prostate displacements in the anteroposterior and superoinferior direction. In comparison with the use of an endorectal balloon, RET placement exhibited to be more favorable as it was able to reduce prostate motion in all directions of about 0–4 mm. Aside from this, the use of an endorectal balloon can cause a distended rectum, which may lead to alteration in the prostate shape and anterior position. Thus, the use of RET can significantly decrease prostate motion following reduced rectal volume changes and it can be more beneficial for patient use than the endorectal balloon [33].

Aside from the rectum, another OAR to be considered in proton beam therapy is the urinary bladder. Takamatsu et al. assessed the benefit of time-fixed bladder control and bladder volume using ultrasonography for prostate cancer patients undergoing proton beam therapy. From March 2011 to September 2013, 75 prostate cancer patients were treated with PBT. Bladder volume was measured prior to irradiation proper at a fixed time of 60 min after urination. Results showed that time-fixed bladder control is closely correlated with bladder volume during treatment, thus increasing the risk of bladder volume inadequacy in the patient. Ultrasonography for prostate cancer patients undergoing proton beam therapy can be beneficial in managing time-fixed bladder control to decrease the probability or toxicity [36].

## 3. Discussion

In 2016, Japanese government commenced providing health care assistance for proton beam therapy in pediatric cancer patients. By 2018, this assistance also expanded catering for prostate cancer patients and head and neck cases. With the high incidence of prostate cancer among Japanese men, increased number of researches in the near future can be expected, which can be used to disseminate the cutting-edge proton therapy in Japan.

Hyogo Ion Beam Medical center had published the most number of articles in this review, using the keywords ‘proton therapy’, ‘prostate cancer’, and ‘Japan’. Most of the equipment for PBT are used not only in Japanese hospitals but also overseas that offer the treatment. These are manufactured by world-renowned Japan based companies such as Hitachi, Toshiba and Mitsubishi Electric. Hence, it can be ascertained that technology transfer of proton therapy to parts of Southeast Asia, beginning next year in Singapore, can be upheld smoothly [37].

This paper focuses on all aspects regarding the usage of proton beam therapy as a treatment for prostate cancer in Japan, including the adverse effects and physical technology. Protons are charged particles that beams stop at a certain depth in a material and produce an energy surge known as Bragg peak at the location of the tumor itself. High-localized deposition of energy can be achieved through the depth-dose profile of the protons that allows an increased radiation dose in tumors while minimizing irradiation to adjacent normal tissues. Dose concentration and escalation without increasing risk is the most basic and important principle in radiation therapy. This property of protons and carbon ions thus accounts to a distinct advantage over other particles such as photons and neutrons [38]. Hence, if organ preservation is prioritized, proton beam can be an ideal therapy for the patient in comparison to other EBRTs. Studies in this literature review include observational, interventional and secondary data analysis. Despite the promising characteristic of proton beam therapy in terms of dose distribution, adverse effects of the treatment are further being studied. One of its most common adverse effects is acute radiation dermatitis (ARD). While a favorable characteristic of proton beam therapy is its ability to lower the radiation dose to the organs at risk, one downside of the modality includes targets that are superficially located. The damage it can cause to the skin can be equal or more pronounced than that of x-ray radiation therapy [39]. Arimura et al. reported on the effectiveness of using film dressing to conceal ARD progressions in comparison to common skin management used [40]. This coincides with a study conducted by Whaley et al., demonstrating protective effect of film dressing towards PBT-induced ARD [41].

Acute and late Gastrointestinal and genitourinary toxicities were also observed in some patients who finished proton beam therapy. Several studies showed less incidence recorded on GI than GU toxicity. Acute GU incidence was noted to be accounted more on the total dose delivered rather than the dose per fraction. Low incidence of GI toxicity associated with PBT is possibly due to the conformation of the lateral beams during dose delivery producing the characteristic dose distribution of proton beams to the target [27]. Grade 3 late GI and GU toxicities were noted to be of very low incidence, not only in the study conducted by Takagi et al, but also from prior results presented by Mendenhall et al. and Bryant et al. Furthermore, late gastrointestinal studies from previous reports of Zelefsky et al., Pollack et al., Michalski et al., Ryu et al., and Slater et al. [42,43,44,45,46,47] were also comparable with the results stated in the study of Nihei et al. [18]. Along with this, heightened toxicity complications were seen on patients who received >70 Gy in conventional PBT [48,49]. There are speculations that indicate that most acute GU symptoms are brought about by urethral mucositis or prostitis since the prescribed selective alpha-1 blocker, which acts primarily on the periurethral smooth muscle, was effective in alleviating its symptoms [50]. Grade 3 GU toxicities may be caused by a poor urinary function prior treatment proper due to the presence of tumor infiltrates in the prostatic urethra causing the organ to be sclerotic [20]. Moreover, patients who have poor baseline urinary functions that were given neoadjuvant AST have a higher probability in experiencing grade 2 GU complications compared to patients who had none [51].

There are also several studies that discuss different factors influencing the accuracy of proton beam irradiation. The movement of the tumor due to respiratory changes was documented in image analysis and may had caused insufficient radiation dose or deficient coverage on the target area leading to inadequacy in regulating the tumor using radiotherapy [52]. In order to surmount this kind of situations, irradiation by gating was put into clinical practice, thus localizing tumors affected by respiratory movement. Bladder volume is also one of the factors affecting accuracy hence, it is usually controlled for the duration of the treatment. PBT is conducted with full bladder volume in order to decrease the probability of toxicity to the organ [30,53,54]. Not only this, it also affects the prostate position, thus, good reproducibility can be achieved through the regulation of bladder volume [36]. Another OAR to be considered is the rectum due to the fact that the volume of this organ can greatly influence the internal motion of the prostate. Study results exhibited that small motions can remarkably reduce the target dose and can slightly be improved by adding sufficient internal margins [55]. As noted by Fuji et al., prostate motion is greatest in the AP direction [33], coherent with reports from Antolak et al. and Van Herk et al., with deviations of 5.1 mm and 2.7 mm, respectively [56,57]. In addition, management of rectum movement can be done with the use of two apparatus namely, the rectum emptying tube or the endorectal balloon. Placing an endorectal balloon is the most typical method to fill the volume of the rectum. McGary et al. documented prostate displacement of <1 mm AP and <3–4 SI mm when endorectal balloon was used [58]. Wachter et al. observed that prostate movement of >5 mm occur in 80% of the patients without the endorectal balloon and only 20% when utilizing the device [59]. However, Van Lin et al. reported no notable difference in the prostate motion with or without the use of an endorectal balloon [60], and thus, the immobilization effect using endorectal balloons is inconsistent. Using these data as comparison, RET reduced the motion of the prostate to within 0–4 mm, and thus can be more favorable than using an endorectal balloon, as reported by Fuji et al. [33].

Photons are the most common beam utilized in EBRT. These particles are devoid of mass and charge and for this reason, it can easily pass through the patient causing energy accumulation as it interacts with excite other electrons. The peak dose takes place a few centimeters upon entering the surface and slowly attenuates as it travels to deeper depths due to photon absorption and dispersion. This event allows a skin-sparing effect; however, the maximal effect upon entrance demonstrates that the highest radiation dose is deposited in the superficial level where normal tissues are affected more than the prostate itself, which is the target organ. These properties of photon particles establish the fact that it is not an optimal form of radiation for targets at great depths within the body. To overcome this, different techniques have been devised such as 3D-CRT, which uses tomography scans for treatment planning as well as IMRT that applies computer algorithms in prescribing doses to the target.

Unlike photons, protons are heavy charged particles which grants positive dosimetric advantages and also has the capacity to avoid extra-target radiation. These features of proton particles allow it to stop within a target given the specific momentum imparted by the acceleration system. At a particular point within the target DNA damage will occur due to the delivered energy by the protons leading to the ionization of molecules. This exceptional property allows protons to exhibit the most impairing effects in the tumor itself [61].

Several dosimetric studies have reported comparison between 3D-CRT, IMRT and PBT in its capacity to spare OAR. Mock et al. documented the potential advantage of PBT and concluded that the main advantage of proton therapy is the significant reduction of the low to medium dose range to the rectum and bladder compared with three-dimensional conformal radiotherapy (3DCRT) or intensity modulated radiotherapy (IMRT) [62]. Likewise, another study compared IMRT and PBT, showed decreased mean rectal and bladder dose by 59% and 35%, accordingly. Table 2 summarizes other studies on the acute and toxicities in photon and proton therapy. Despite the akin incidences of late GI morbidities due to PBT and IMRT, there are more favorable outcomes in late GU toxicities after PBT than those of IMRT due to the reduction of dosage radiation enclosing the OARs. In fact, higher radiation dose to the rectum have shown significant association in the development of late GI complications while broader dose to the bladder showed pertinent relation in developing GU toxicities [63]. Regardless of the more pronounced skin injury caused by PBT, it is a more superior choice of modality in treating prostate cancer.

The overall five-year biochemical relapse free rates (bRFS) recorded by authors included in this literature review coincide with other international retrospective studies [25,54,64] and patients who used proton beam irradiation were also documented to have higher overall bRFS rates compared to patients given neoadjuvant and adjuvant long term ADT. In this manner, implying a more beneficial outcome using PBT than hormonal therapies. A comparison study also observed a risk-reduction of about 26–39% for secondary malignancy risk for PBT patients compared to those who underwent IMRT [65]. As the spot scanning method can reduce neutron scatter compared with passive scattering, proton therapy using this technique may present a positive treatment outcome for younger patients. However, as shown in the prognostic factors analysis, younger patients tend to experience more biochemical relapse and less late toxicity; personalized treatment in patient age should be considered [25].

Published journals in Japan on proton beam do not only focus on the adverse effects of PBT but also includes interventional studies to further improve PBT such as in physical technology and delivery system to increase favorable outcomes and quality of life for the patients. Treatment planning system is utilized in clinical planning routinely, and thus, computation times, as well as accuracy, are very important factors to consider in devising one. At present, the generally used algorithms for proton dose calculations are pencil beam algorithms [66]. However, Monte Carlo method is an exceptional approach which provides the highest accuracy in compared to other analytical models, yet, due to its extensive computation time of about six hours and 2% uncertainty in <1 minute each patient, it is not commonly used in clinical practice [67]. With this, many fast Monte Carlo approaches have been formulated and documented including the fast dose calculator track-repeating algorithm by Yepes et al., simplified Monte Carlo with fewer physics processes by Kohno et al., and Hotta et al. [66,67,68,69]. These approaches were adapted in clinical implementations and can be applied in clinical proton treatment planning.

Patients of proton beam therapy for prostate cancer had been continuously growing in number due to its promising characteristics of high dose distribution in the target and a sharp distal fall-off. Treatment of tumors using proton irradiation uses beam delivery system- either the beam scattering or beam scanning methods. The more conventional method is the beam scattering, which utilizes a passive beam delivery system consisting of a modulator, collimator, and compensator to achieve proper dosage for target shape. With the use of a ridge filter, a spread-out Bragg peak can be created, conforming to the size of the target volume. The beam scattering method can deliver homogenous dosage along the target volume. However, one problem of this method is the accumulation of high doses in some of the proximal parts of the target affecting the normal tissues. In contrast, the beam scanning method has an active delivery system that can move the peak position in a robust manner within the target by altering beam energy and/or beam penetration with the use of observers. With these, it can deliver precise and sufficient dosage that corresponds to the target volume [30]. The use of pencil beam, a type of beam scanning method, for treatment of PCa can be more favorable considering the close proximity of the prostate to the organs at risk, hence, lessening the dose delivery to these parts. In addition, a better, more adaptable dose distribution and reduced preparation time before starting the treatment proper can be carried out, thus allowing better tumor control rate and decreased risk of normal tissue toxicity and development of radiation induced secondary cancer.

Collimators are used as a part of the passive scattering system. The conventional circular collimator controls the beam size with 10-diameter choices from 0.5 to 6 cm allowing the system to generate a sharp dose drop-off as well as a low dose to the organs at risk. The only disadvantage of the passive scattering system with circular collimators is its tedious and time-consuming process due to the demand of large quantity monitor units in order to deliver the desired dose. The development of multileaf collimators enable the fields to freely complement the tumor shape, provide a more efficient dose delivery resulting to better dose distribution, reduce monitor units, and finally lessen treatment times on different cases including prostate cancer patients [31,70,71]. In 2015, McGuiness et al. designed a study plan suggesting a more homogenous and shorter treatment time using MLC. The target coverage and conformity of the study plan corresponds with the results of Murai et al., despite the difference in prescribed dose and fractionation numbers [30,70]. Another material used in proton therapy are fiducial gold markers which are used in treating organs displaying motion. It is a very beneficial technique that allows higher accuracy in delivering dose to moving targets and decreasing irradiation to adjacent normal tissues. Nonetheless, this still poses the disadvantage of causing an under-dose effect due to the high electron density of the element. With this, multifield irradiation has been documented to overcome this problem depending on the size of the marker. A patient embedded with a 1.5 mm diameter gold fiducial marker can surmount the under-dose effect and conceal tumor control probability reduction with the use of two or more fields during proton beam therapy. For 2.0 mm diameter markers, it is documented that more than two fields should be used to balance out the effect of using the markers.

## 4. Materials and Methods

Published work on Proton Beam Therapy for Prostate Cancer in Japan were gathered using PubMed database in the second-half of 2018. We went through abstracts and manuscripts written in English with the keywords ‘proton beam therapy’, ‘prostate cancer’ and ‘Japan’. Research details were as follows (“proton therapy”[MeSH Terms] OR (“proton”[All Fields] AND “therapy”[All Fields]) OR “proton therapy”[All Fields] OR (“proton”[All Fields] AND “beam”[All Fields] AND “therapy”[All Fields]) OR “proton beam therapy”[All Fields]) AND (“prostatic neoplasms”[MeSH Terms] OR (“prostate”[All Fields] AND “cancer”[All Fields]) OR “prostate cancer”[All Fields]) AND (“japan”[MeSH Terms] OR “japan”[All Fields]). In this literature review, everything that discussed proton beam therapy for prostate cancer in Japan were included. This identified 23 articles. Analysis of journals included in this study is shown in Figure 1 using a flowchart by PRISMA [72].

Journals in this study encompasses observational, interventional and secondary analysis of proton therapy for prostate cancer in Japan. There were a total of 50 articles and exclusion also applied to journals such as: (i) Not written in English *n* = 5. (ii) focused on other radiation therapy apart from proton beam *n* = 8. (iii) hospitals and subjects were not situated in Japan *n* = 4. (iv) focused on other type of cancer apart from prostate neoplasm *n* = 4. (v) Full texts not available online *n* = 6.

Applying these, 23 journals were eligible for this literature review. No delimitation for the year published was applied.

## 5. Conclusions

Currently there are 14 proton beam facilities all over Japan, four of which belongs to Japanese national universities. The use of proton beam therapy had not only caught the interest of the physicians, but also researchers working at national universities. Hence, interdisciplinary research is expected to proceed in diversified fields such as engineering, science, and sociology. In this period, there have been several journals published regarding proton beam irradiation, yet it is anticipated to continuously increase in number due to its progressing popularity as a promising treatment for localized prostate cancer.

In 2016, pediatric patients were the first subjects wherein proton beam therapy was covered by Japanese universal health care. Starting April 2018, proton beam therapy for prostate cancer was also included in the Japanese NHI coverage. From these mandates, it is predicted that different cancer site cases will become available for proton therapy and more Japanese individuals can benefit from it.

There have been various studies regarding the efficacy and adverse effects of patients who are underwent proton beam therapy for localized prostate cancer. The incidence of acute and late toxicities concerning the GI and GU tract, as well as the radiation doses to the OARs are significantly lower compared to other external beam radiation therapy (EBRT). Proton therapy monotherapy and multimodal therapy had also shown its benefits in patients’ prognosis and quality of life. Biochemical control of patients who completed proton therapy are significantly favorable in PCa patients, including those with high and very high risk cases. Despite the favorable results of proton beam therapy shown in these journals, further research should include more patients and longer follow-up schedules to clarify the definitive role of PBT. Nonetheless, up to recent retrospective studies included in this paper concluded that PBT can be a suitable treatment option for localized prostate cancer.

The development of engineering methods which enables humans to easily control proton beams is one of the research fields directly linked to the enhancement of medicine. It can be considered that Japan is one if the leading countries in equipment related procedures regarding proton beam therapy, including heavy particles due to the fact that Japan comes first in engineering-related technology in the world.

Lastly, publications from Japanese researchers on proton beam therapy for prostate cancer in journals with varying impact factors may also help medical and research institutions. These can significantly aid in distinguishing the treatment planning, treatment proper, benefits, and advancements in the country that can be more likely applied and adapted to institutions that utilizes Japanese equipment.

## Figures and Tables

**Figure 1 jcm-08-00048-f001:**
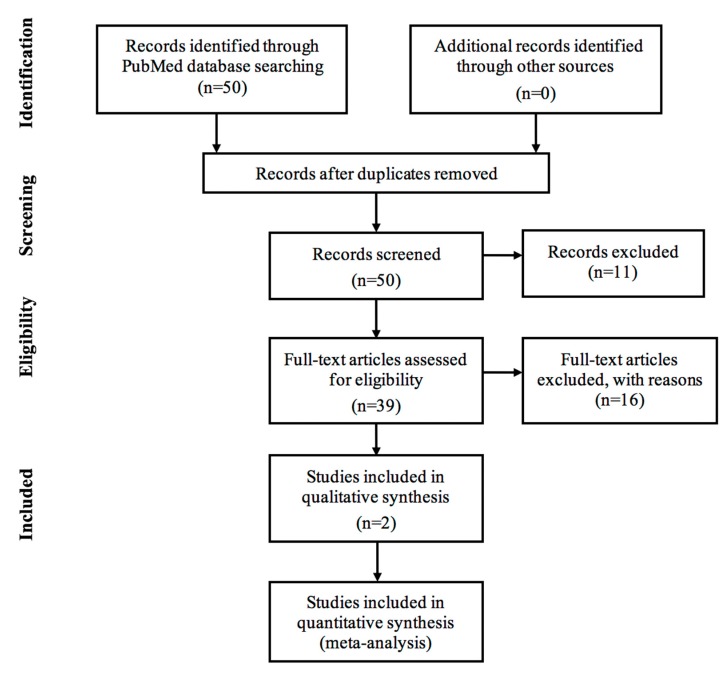
Analysis of the journals flow diagram [72].

**Table 1 jcm-08-00048-t001:** Article details.

Study Type	Hospital	Year Pulished	Journal (Impact Factor)	Article Title	Author/s	Number of Patients	Treatment Planning Equipment (Proton Beam Irradiation Equipment)
**Observational Study**							
	Medipolis Proton Therapy and Research Center	2018	Cancers (5.326)	Proton Beam Therapy Alone for Intermediate- or High-Risk Prostate Cancer: An Institutional Prospective Cohort Study	Takeshi Arimura et al.	218	XiO-M CT based 3D treatment planning system, Elekta, Stockholm, Sweden
	Fukui Prefectural Hospital	2018	Medical Physics (2.617)	Effects of organ motion on proton prostate treatments, as determined from analysis of daily CT imaging for patient positioning	Yoshikazu Maeda et al.	10	XiO-N Proton Treatment Planning System (Elekta Corp., Stockholm, Sweden)
	Fukui Prefectural Hospital	2018	Medical Physics (2.617)	Positioning accuracy and daily dose assessment for prostate cancer treatment using in-room CT image guidance at a proton therapy facility	Yoshikazu Maeda et al.	30	XiO-N Proton Treatment Planning System (Elekta Corp., Stockholm, Sweden)
	National Cancer Center Hospital East, Shizuoka Cancer Center, Hyogo Ion Beam Medical Center, University of Tsukuba Faculty of Medicine, Southern TOHOKU Proton Therapy Center, Fukui Prefectural Hospital, Medipolis Proton Therapy and Research Center	2018	Cancer Medicine (3.362)	Long term outcomes of proton therapy for prostate cancer in Japan: a multi-institutional survey of the Japanese Radiation Oncology Study Group	Hiromitsu Iwata et al.	1291	FOCUS-M, CMS, St. Louis MO, Mitsubishi Electric Corp (Kobe, Japan) and VQA, Hitachi Ltd. (Tokyo, Japan),
	University of Tsukuba Department of Radiation Oncology and Proton Medical Research Center	2017	Molecular and Clinical Oncology	A retrospective study of late adverse events in proton beam therapy for prostate cancer	Hirokazu Makishima et al.	93	Hitachi 3D Treatment Planning System ver. 2.0, Hitachi Ltd. (Tokyo, Japan)
	Hyogo Ion Beam Medical Center	2017	Cancer Medicine (3.362)	Long-term outcomes in patients treated with proton therapy for localized prostate cancer	Masaru Takagi et al.	1375	Not Mentioned
	Southern Tohoku Proton Therapy Center	2017	Physica Medica (2.24)	Effect of DIR on prostate passive-scattering proton therapy dose accumulation	Yoshimoto Abe et al.	10	Proton Therapy System (Mitsubushi Electric, Kobe, Japan) Aquilion LB Treatment Planning CT simulation (Toshiba Medical Systems, Tokyo, Japan) Xio-M System PSPT treatment planning (Elekta, Stockholm, Sweden)
	Hokkaido University Hospital	2017	Journal of Radiation Research (2.031)	A simulation study on the dosimetric benefit of real-time motion compensation in spot-scanning proton therapy for prostate	Yusuke Fujii et al.	9	VQA, Hitachi Ltd. (Tokyo, Japan)
	Shizuoka Cancer Center	2012	Journal of Radiation Research (2.031)	A Treatment Planning Comparison of Passive Scattering and Intensity-Modulated Proton Therapy for Typical Tumor Sites	Yuki Kase et al.	16 (4 PCa Patients)	Xio-M Elekta CMS Software (Elekta, Stockholm, Sweden and Mitsubishi Electric Corp., (Kobe, Japan) *
	National Cancer Center Hospital East, Shizuoka Cancer Center, Hyogo Ion Beam Medical Center	2011	International Journal of Radiation Oncology BiologyPhysics (5.554)	Multi-institutional Phase II Study of Proton Beam Therapy for Organ-confined Prostate Cancer Focusing on the Incidence of Late Rectal Toxicities	Keiji Nihei et al.	151	Not Mentioned
	Hyogo Ion Beam Medical Center	2009	International Journal of Radiation Oncology Biology Physics (5.554)	Physiologic Reactions after proton Beam Therapy in Patients with Prostate Cancer: Significance of Urinary Autoactivation	Masakazu Shimizu et al.	59	FOCUS-M, CMS Japan (Tokyo, Japan) and Mitsubishi Electric Corp. (Kobe, Japan)
	Hyogo Ion Beam Medical Center	2007	International Journal of Radiation Oncology Biology Physics (5.554)	Acute Morbidity of Proton Therapy for Prostate Cancer: The Hyogo Ion Beam Medical Center Experience	Hiroshi Mayahara et al.	287	FOCUS-M, CMS Japan (Tokyo, Japan) and Mitsubishi Electric Corp. (Kobe, Japan)
	National Cancer Center Hospital East	2005	Japanese Journal of Clinical Oncology (2.370)	Phase II Feasibility Study of High-Dose Radiotherapy for Prostate Cancer Using Proton Boost Therapy: First Clinical Trial of Proton Beam Therapy for Prostate Cancer in Japan	Keiji Nihei et al.	30	-
**Interventional Study**							
	Nagoya City Hospital	2018	International Journal of Clinical Oncology (2.61)	Acute toxicity of image-guided hypofractionated proton therapy for localized prostate cancer	Koichiro Nakajima et al.	526	VQA Hitachi (PROBEAT III: Hitachi Ltd., Tokyo, Japan)
	Nagoya Proton Therapy Center	2017	Journal of Radiation Research (2.013)	Comparison of multileaf collimator and conventional circular collimator systems in Cyberknife stereostatic radiotherapy	Taro Murai et al.	10	Multiplan treatment planning system ver 5.1 (Accuray Inc., Sunnyvale, CA, USA)
	Medipolis Proton Therapy and Research Center	2016	International Journal of Radiation Oncology Biology Physics (5.554)	Effects of Film Dressing on Acute Radiation Dermatitis Secondary to Proton Beam Therapy	Takeshi Arimura et al.	271	Not Mentioned
	Medipolis Proton Therapy and Reseach Center	2015	Targeted Oncology (3.877)	HPMA Copolymer-Conjugated Pirarubicin in Multimodal Treatment of a Patient with Stage IV Prostate Cancer and Extensive Lung and Bone Metastases	Haruhiko Dozono et al.	1	XiO-M: CMS and Mitsubishi Electric Corp. (Tokyo, Japan)
	Fukui Prefectural Hospital	2014	Japanese Journal of Radiology (1.044)	Utility an initial adaptive bladder volume control with ultrasonography for proton-beam irradiation for prostate	Shigeyuki Takamatsu et al.	75	XiO-N: ELEKTA, Stockholm, Sweden & Mitsubishi Electric Corp., Kobe, Japan
	Hokkaido University Hospital	2012	Medical Physics (2.617)	Biological effect of dose distortion by fiducial markers in spot scanning proton therapy with a limited number of fields: A simulation study	Taeko Matsuura et al.	10	Hitachi (Hitachi Ltd., Tokyo, Japan)
	National Cancer Center Hospital East	2011	Physics in Medicine and Biology (2.811)	Clinical implementation of a GPU-based simplified Monte Carlo method for a treatment planning system of proton beam therapy	R Kohno et al.	4	Not Mentioned
	Shizuoka Cancer Center	2009	Japanese Journal of Radiology (1.044)	Changes in rectal volume and prostate localization due to placement of a rectum-emptying tube	Hiroshi Fuji et al.	21	Asterion Treatment planning system (Toshiba Medical Systems, Tokyo, Japan) XiO-M (Mitsubishi Electric Corp., Tokyo, Japan)
**Secondary Data Analysis**							
		2016	Japanese Journal of Clinical Oncology (2.370)	Proton beam therapy in Japan: current and future status	Hideyaki Sakurai et al.		N/A
		2015	International Journal of Urology (1.884)	Particle radiotherapy for prostate cancer	Yoshiyuki Shioyama et al.		N/A

Abbreviation: * Mitsubishi Electric Corp. (Kobe, Japan) extended Xio-M for PSPT use.

**Table 2 jcm-08-00048-t002:** Summarized reports on radiotherapy for PCa.

Author	Year	EBRT	Patients	Dose				Acute Toxicity					Late Toxicity					5-Year Biochemical Relapse Free Survival	
				(GyE)		Genitourinary (Number)			Gastrointestinal			Genitourinary			Gastrointestinal				
					Grade 1	Grade 2	Grade 3	Grade 1	Grade 2	Grade 3	Grade 1	Grade 2	Grade 3	Grade 1	Grade 2	Grade 3	Low Risk	Intermediate Risk	High Risk
Nihei et al. [18]	2005	Photon + Proton	30	50Gy + 26	66.7% (20)	13.3% (4)	NR	56.7% (17)	0%	0%	6.67% (2)	10% (3)	NR	26.67% (8)	10% (3)	NR		NR	
Mayahara et al. [20]	2007	Proton	287	60	54% (154)	39% (111)	1.1% (4)		NR			NR			NR			NR	
Kupelian et al. [21]	2007	IMRT	770	70		NR			NR		NR	7% (36)	NR	NR	6% (30)	NR	94%%	83%	72%
Cahlon et al. [22]	2008	IMRT	478	86.4	NR	22% (105)	0.6% (3)	NR	8%(37)	0%	NR	13% (60)	<3% (12)	NR	3% (16)	<1% (2)	98%	85%	70%
Nihei et al. [17]	2011	Proton	151	74	NR	12.7% (9)	0%		NR		NR	6%	0%		NR			NR	
Spratt et al. [23]	2013	IMRT	1002	86.4		NR			NR		NR	21.1% (211)	2.2%(22)	NR	4.4% (44)	0.7% (7)	98.8% *	85.6 *	67.9 *
Nakajima et al. [24]	2017	Proton	526																
		CPT	254	74 (LR)	NR	15% (38)	NR	0.8% (2)	NR	NR		NR			NR			NR	
				78 (HR)															
		HFPT	272	60 (LR)		5.9% (16)		0.7% (2)				NR			NR			NR	
				63 (HR)															
Takagi et al. [25]	2017	Proton	1375	74		NR			NR		8.7% (119)	2.4% (33)	0.07% (1)	6% (82)	4% (53)	0.07% (1)	99%	91%	86%
Makishima et al. [9]	2017	Proton	93	74 (LR)		NR			NR		0%	5.4% (5)	1.08% (1)	NR	4.3% (4)	NR	NR	99%	NR
				78 (IR, HR)															
Iwata et al. [26]	2018	Proton	1291	70-80		NR			NR			4.1% (53)	0.5% (6)		4.0% (52)	0.3% (4)	97%	91.10%	83.10%
Arimura et al. [27]	2018	Proton	218	70		13.1%%						0%			0.00%		NR	97%	83%
				74	NR	27.90%	NR		NR		NR	2.3% (5)	NR	NR	2.30%	NR			
				78		28.10%						10.5% (23)			10.50%				

Abbreviations: LR, low risk; IR, intermediate risk; HR, high risk; NR, no record; * 7-year biochemical relapse free survival.

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
