# Peer review of "A Literature Review of Proton Beam Therapy for Prostate Cancer in Japan"

_jcm, 2019, doi:10.3390/jcm8010048_

Round 1

Reviewer 1 Report

The authors present a review article considering the potential benefit of proton therapy in the treatment of prostate cancer. This review is based on the significant proton therapy presence in Japan. 

In general the concept of the review is relevant and current, however, execution of the work lets the manuscript down. From a grammatical perspective the whole review requires careful English grammar editing. 

In the first half of the review, the authors compare incidence rates of various parameters (e.g. acute toxicity, late toxicity, biochemical relapse etc...), however, too often these end up as a long list of percentages, which do not clearly identify the specific comparators.

With respect to the treatment planning section, this is a critical component of any proton/photon treatment, however, the content provided in this section is based entirely on one paper [ref 10]. This section should be expanded upon, using a wide range of references. With respect to the toxicity section, on some occasions the authors attempted to draw comparisons between the more commonly used photon therapy and proton therapy. In many of these instances the comparisons were poorly made, and there was not clear statement of superiority of one approach over the other. 

Table 1 adds very little to the manuscript. This is little more than a list of references that are already included in the bibliography. I would suggest that this table, rather than having the year published and title of the journal, should have columns of the primary and secondary measure of treatment efficacy, and details of risk rates. If the data were accessible this should also include comparisons of outcomes between proton/proton therapies. It's not clear why the authors have included a list of the journal impact factors and frequency of publication. 

In many instances, particularly in pages 10 - 12 the authors details occurrence rates of various treatment related effects, however, there is very little effort made to explain the importance/significance of the occurrences. Slightly more attention should be given to impact. 

The second half of the review focus on the physical technology and the means of delivery. While this is all relevant, the authors should make it clear from the offset that the review has two distinctive components. This should be provided upfront. 

Many of the abbreviations used are not first defined. Check and amend this through the entire manuscript. Also, the information provided in lines 95-97 cannot be extrapolated from figure 1. Finally figure 2 adds nothing addition to the manuscript, which is not already provided in the text.

Author Response

REVIEWER 1

Point 1: In the first half of the review, the authors compare incidence rates of various parameters (e.g. acute toxicity, late toxicity, biochemical relapse etc...), however, too often these end up as a long list of percentages, which do not clearly identify the specific comparators.

Response 1:Thank you for your comment. In line with this, the authors have created a table  (Table 2) on the adverse effects and biochemical relapse allowing the readers to easily identify the difference between proton therapy and other external beam radiation therapies in accordance with these factors. 

Point 2: With respect to the treatment planning section, this is a critical component of any proton/photon treatment, however, the content provided in this section is based entirely on one paper [ref 10]. This section should be expanded upon, using a wide range of references. With respect to the toxicity section, on some occasions the authors attempted to draw comparisons between the more commonly used photon therapy and proton therapy. In many of these instances the comparisons were poorly made, and there was not clear statement of superiority of one approach over the other. 

Response 2:Thank you for your comment. We agree that the treatment planning section is indeed one of the most critical aspects of proton treatment. With this, we added other references to expound the process and hopefully to discuss more about this section. As for the toxicity, we added a paragraph comparing photon and proton therapy in the discussion to corroborate the  comparison between the two therapies. 

Point 3: Table 1 adds very little to the manuscript. This is little more than a list of references that are already included in the bibliography. I would suggest that this table, rather than having the year published and title of the journal, should have columns of the primary and secondary measure of treatment efficacy, and details of risk rates. If the data were accessible this should also include comparisons of outcomes between proton/proton therapies. It's not clear why the authors have included a list of the journal impact factors and frequency of publication. 

Response 3:Thank you for your comment. We added this table to show the inclusive journals in this review used in order to avoid confusion on which are only used to strengthen the claims of the 23 articles in the review. Former table 2 was revised and now shows comparison between data on photon and proton therapies. We have included the journal impact factors and frequency of publication to demonstrate the significance of these articles by the impact factor of the journals from which it were published. 

Point 4: In many instances, particularly in pages 10 - 12 the authors details occurrence rates of various treatment related effects, however, there is very little effort made to explain the importance/significance of the occurrences. Slightly more attention should be given to impact. 

Response 4:Thank you for prudent comment. In order to add impact on the rates presented as well as the adverse effects, we added couple of sentences to present and corroborate the significance of the rates provided. Also, we explained the features of the toxicities which can help in identifying or grading the complication experienced by the patient.

Point 5:The second half of the review focus on the physical technology and the means of delivery. While this is all relevant, the authors should make it clear from the offset that the review has two distinctive components. This should be provided upfront. 

Response 5:Thank you for your comment. The whole review focuses on proton beam therapy for prostate cancer thus, these aspects are also included. To make it apparent, we added a sentence that states these will also be discussed in the review.

Point 6: Many of the abbreviations used are not first defined. Check and amend this through the entire manuscript. Also, the information provided in lines 95-97 cannot be extrapolated from figure 1. Finally figure 2 adds nothing addition to the manuscript, which is not already provided in the text.

Response 6:Thank you for pointing this out. It was an author’s mistake. The abbreviations have now been double-checked and Figure 2 have been removed to avoid redundancy. 

Reviewer 2 Report

The literature review is extensive and several points should be revised, e.g.:

Readability is  unsatisafactory

Several parts are not useful for the review and should be deleted or shortened (i.e table 2, the enitre section 2 etc...)

The review layout should be revised (ie position ofsection 4, materials and methods should be changed)

Author Response

REVIEWER 2

Point 1: The literature review is extensive and several points should be revised, e.g.:

Readability is  unsatisafactory

Point 2: Several parts are not useful for the review and should be deleted or shortened (i.e table 2, the enitre section 2 etc...)

Response for 1 and 2:Thank you for your comment. We’re sorry for giving you a hard time on reading the manuscript. We have went over the whole article again, revised the wordings and deleted sentences which makes the whole context unclear. Table 2 was revised into a summary of photon and proton therapy, while section 2 was shortened since most of the percentages were transferred to the table. 

Point 3: The review layout should be revised (ie position ofsection 4, materials and methods should be changed)

Response 3:Thank you for pointing this out. The authors have followed the template given by Journal of Clinical Medicine (MDPI Format), however, if this should be followed, we will gladly change its organization.

Reviewer 3 Report

Dr. Rika Hoshina and collegues write a  literature review based on 23 published articles related to proton beam therapy (PBT) for prostate cancer in Japan. The article mentiones observational studies,  interventional studies or secondary data analysis. 

Only some considerations are reported below.

-In the introduction section the authors well describe the prostate structure  and the panorama of the recent and actual solutions for PCa in which their consierations .

However, they don't mention some solutions, as for example androgen-deprivation therapy (with its pro and con) and other recent  therapies as photodynamic therapy (PDT). The multimodal

treatment that involves different therapeutic agents acting synergistically and/or in combination with techniques has emerged as a successful approach. In this context, light-triggered therapies in localized and partially metastasized PC should be mentioned considering recent advances: 

1) doi: 10.1021/acs.bioconjchem.5b00261 and 

2) PMID: 20882130;

-check "75 proton beam therapy for prostate cancer and head and neck cancer" font  line 75

The reviews should also clarify better some questions that in part are addressed in the manuscript, probably adding a summary table or discussing when it is possible. The questions are:

- All men can be treated with Proton Beam Therapy? Every patient with a particular and definite clinical picture will receive the same doses of PBT? 

- A table resuming risks and benefits is encouraged;

- Is there any special preparation for patients needed before the procedure?

-What are the most particular symptoms during and after PBT procedure?

- Resume in a table the relative benefits and indications for PBT, please.

Author Response

REVIEWER 3

Point 1: However, they don't mention some solutions, as for example androgen-deprivation therapy (with its pro and con) and other recent  therapies as photodynamic therapy (PDT). The multimodal

treatment that involves different therapeutic agents acting synergistically and/or in combination with techniques has emerged as a successful approach. In this context, light-triggered therapies in localized and partially metastasized PC should be mentioned considering recent advances: 

1) doi: 10.1021/acs.bioconjchem.5b00261 and 

2) PMID: 20882130;

Response 1:Thank you for prudent comment. We are very grateful for providing us these articles to be included in our literature review. We had added a paragraph which included other modalities used in proton therapy. Some multi-modal approach studies were also discussed under the section of Acute Toxicity and Biochemical Relapse and survival rate. 

Point 2: -check "75 proton beam therapy for prostate cancer and head and neck cancer" font  line 75

Response 2:Thank you for pointing this out. It was an author’s mistake. We have edited the particular lines, as well as other typos in the article. 

Point 3: The reviews should also clarify better some questions that in part are addressed in the manuscript, probably adding a summary table or discussing when it is possible. The questions are:

- All men can be treated with Proton Beam Therapy? Every patient with a particular and definite clinical picture will receive the same doses of PBT? 

- A table resuming risks and benefits is encouraged;

- Is there any special preparation for patients needed before the procedure?

-What are the most particular symptoms during and after PBT procedure?

- Resume in a table the relative benefits and indications for PBT, please.

Response 3:Thank you for your comment and also the questions which acted as a guide to improve this literature review. We’ve mentioned that each case has an individualized treatment so not all will receive the same proton dose. No special preparations are needed, but this may depend on the facility which will conduct the therapy. PBT can cause acute injuries as mentioned such as radiation dermatitis, urinary autoactivation, genitourinary and gastrointestinal toxicities. But this are relative events, and does not happen to all patients who are undergoing or underwent PBT.  For the risks, benefits and indications, in lieu of proving a table, we provided a paragraph to discuss these aspects. 

Round 2

Reviewer 1 Report

I am happy that the manuscript be published in the current form.